# Development and Validation of the Knowledge of Human Papillomavirus Scale in Japan

**DOI:** 10.3390/healthcare13192536

**Published:** 2025-10-08

**Authors:** Ayano Tokuda, Atsuko Shiota, Pasang Wangmo, Kimiko Kawata

**Affiliations:** 1Kagawa University Hospital, Miki 761-0793, Kagawa, Japan; tokuda.ayano.zy@kagawa-u.ac.jp; 2School of Nursing, Faculty of Medicine, Kagawa University, Miki 761-0793, Kagawa, Japan; shiota.atsuko@kagawa-u.ac.jp; 3School of Nursing, Graduate School of Medicine, Kagawa University, Miki 761-0793, Kagawa, Japan; s24d751@kagawa-u.ac.jp

**Keywords:** HPV vaccines, papillomavirus infections, reproducibility of results, HPV knowledge scale, Japan

## Abstract

**Background/Objectives**: In Japan, the human papillomavirus (HPV) vaccine introduction process is unique, and no HPV knowledge scale with established reliability and validity currently exists. This study aimed to develop a new HPV knowledge scale and evaluate its reliability and validity for practical use. **Methods**: With permission from the original authors of the HPV Knowledge Scale (Jo Waller et al.), we created a Japanese version incorporating the original two subscales and adding new items. The translation process involved multiple researchers, back-translation by a professional agency, and expert review to ensure linguistic and contextual accuracy. The study was approved by the Clinical Research Ethics Review Board of the researchers’ affiliated universities and conducted between April and August 2024. **Results**: Reliability and validity were assessed using data from 793 parents of junior high school students, including both boys and girls. Confirmatory factor analysis showed a good model fit (Goodness-of-Fit Index [GFI] = 0.934, Adjusted GFI [AGFI] = 0.907, Comparative Fit Index [CFI] = 0.928, Root Mean Square Error of Approximation [RMSEA] = 0.063). Cronbach’s alpha ranged from 0.688 to 0.845 and item-total correlations ranged from 0.393 to 0.584. Test–retest reliability, assessed with Spearman’s rank correlation, was r = 0.791 (*p* < 0.001). The final scale, named the Japan HPV Knowledge Scale (J-HPV-KS), includes 17 items across five factors. **Conclusions**: The J-HPV-KS covers HPV-related diseases, transmission routes, natural history, and vaccines. It demonstrated sufficient reliability and validity for use in Japan and is a useful tool for assessing HPV-related knowledge among Japanese parents and guardians.

## 1. Introduction

The human papillomavirus (HPV) vaccination rate in Japan remains significantly lower than in other developed countries, making improved coverage an urgent public health concern. For example, high vaccination coverage has been reported in Norway and Portugal, with similar trends observed in other countries such as Sweden and Denmark [1]. According to a recent systematic review, HPV vaccination coverage data were available from 132 countries, and the estimated global first-dose coverage was 61.6% (95% CI: 50.9–71.8%). HPV vaccination coverage was estimated to be highest in the SDG regions of Oceania, at 69.2% (95% CI: 57.8–70.7%), and lowest in Eastern and South-Eastern Asia, at 31.6% (95% CI: 19.1–57.5%) [2].

In Japan, the HPV vaccine was approved in 2009. Routine, publicly funded HPV immunization was introduced in 2013; however, media reports of adverse events such as pain and motor dysfunction prompted the Ministry of Health, Labor and Welfare to suspend its active promotion of the vaccine a few months later [3,4]. This decision led to a drastic decline in vaccination rates from an initial 70% at the start of the program to less than 1% [5]. Subsequent reviews by the Health Sciences Council, based on domestic and international studies, found no clear evidence of a causal relationship between the reported adverse events and the HPV vaccine [4]. Consequently, proactive recommendations for routine vaccination of girls aged 12 to 16 years resumed in April 2022 [6]. Nevertheless, in 2022, the HPV vaccination rate for the first dose remained at just 42.2%, which is still inadequate [7]. This percentage is considered inadequate as it falls below the WHO target of at least 90% HPV vaccination coverage among girls by age 15, which is one of the 90–70–90 targets that must be met by 2030 for countries globally to be on the path towards cervical cancer elimination [8]. A modeling study in China [9] demonstrated that achieving the 90–70–90 targets would reduce cervical cancer incidence to fewer than 4 cases per 100,000 women by 2093, illustrating how reaching these targets can bring substantial benefits to populations worldwide.

Women vaccinated at 14–22 years of age and given three doses of the bivalent vaccine showed a significant reduction in incidence compared with all unvaccinated women, although this finding is based on data from the bivalent vaccine [10]. Accordingly, there has been progress in recommending HPV vaccination for both males and females internationally. In the United States, reports suggest that more than 90% of cervical and anal cancers, and 63–75% of penile, vulvar, vaginal, and oropharyngeal cancers are caused by HPV [11,12]. Another study suggests that the acquisition of HPV is common among males, with patterns similar to those in females [13]. A systematic review also supports early vaccination of boys before the onset of sexual activity [14]. Among the 148 WHO member states, 72 (49%) limited vaccination to females, whereas the remaining 76 (51%) implemented vaccination programs for both sexes as of 2024 [2]. Twenty-six of the 27 European Union (EU) Member States, as well as Norway and Iceland, provide HPV vaccinations for adolescents as part of the national program, targeting children aged 12–13 [15,16,17]. Vaccinating males helps prevent conditions such as genital warts, anal cancer, and oropharyngeal cancer, and also contributes to herd immunity, thereby reducing HPV infection rates among females.

Regarding Japan, routine HPV vaccination is provided free of charge for girls from the first day of the school year (1 April) in which they reach the age of 13 (equivalent to the first year of junior high school) until the end of the school year in which they reach the age of 16 (equivalent to the first year of senior high school). For boys, since discussions on routine vaccination have only recently begun in Japan, vaccination is voluntary and the target age is 9 years and older (with no upper age limit); however, in municipalities that provide public subsidies, the target age range is the same as that for girls. The high cost—approximately 20,000 yen per dose, or 50,000–60,000 yen if the full three-dose series is completed (about 330–400 USD)—acts as a barrier, limiting the number of individuals seeking vaccination [18]. Several European countries that introduced HPV vaccination early have reported declines in the incidence of HPV-related diseases. The Australia was the first country to introduce a national HPV vaccination program for girls in 2007, which was later expanded to include boys in 2013 [19]. Consequently, a study comparing trends in the proportions of new genital warts diagnoses in Australia between the pre-vaccination and vaccination eras reported a marked reduction [20]. Furthermore, HPV vaccination recommendations are being expanded to include special populations, such as childhood cancer survivors [21], men who have sex with men, and immunocompromised individuals [22].

In Japan, HPV vaccination behavior is influenced by the Immunization Act, which requires parental consent for individuals under 15 years old to be vaccinated [23]. As a result, parental decision-making plays a significant role in vaccine uptake. Previous studies in Japan have shown that approximately 90% of junior high school girls who received the HPV vaccine cited their mothers’ opinions as the most influential factor in their decision [24], highlighting the strong impact of parental influence. Additionally, parental explanations about HPV vaccination have been identified as key factors associated with children’s willingness to receive the vaccine [25]. Although target populations and vaccination rates vary across countries and regions, global studies have examined the factors related to parents’ willingness to vaccinate. A systematic review and meta-analysis of 79 studies from 15 countries reported that factors positively influencing parents’ decisions included general vaccine beliefs, perceived benefits, awareness of risks from non-vaccination, and knowledge of HPV and cervical cancer [26]. Other studies have also found associations between mothers’ knowledge of HPV infection and vaccination [27,28,29,30], as well as their knowledge of cervical cancer [28], and their willingness to vaccinate their children. Conversely, low levels of knowledge have been linked to prejudice and hesitancy regarding vaccination [31,32]. However, some studies have reported no significant association between parental knowledge and actual vaccination uptake among adolescents [33]. Notably, the definitions and measures of HPV-related knowledge varies across these studies.

Providing parents with accurate information about the benefits of the vaccine may facilitate informed decision-making and encourage initiation and completion of the full HPV vaccine series [34]. In Japan, although various studies have examined factors influencing parental decisions, most were conducted before the reinstatement of proactive vaccination, and there is limited research on decision-making and knowledge dissemination since then. Japan’s unique history with HPV vaccine introduction may affect how parents acquire and interpret information, highlighting the need for a scientifically validated scale to measure HPV vaccine knowledge. However, no validated, context-specific scale currently exists in Japan to assess the knowledge of vaccine recipients themselves as well as their parents, despite their critical role in HPV vaccination decision-making. Understanding current knowledge levels and identifying information gaps are essential for designing effective educational initiatives to increase HPV vaccination rates. Therefore, we developed an HPV knowledge scale and evaluated its reliability and validity to support practical application in public health and education.

## 2. Materials and Methods

### 2.1. Selection of Items for the HPV Knowledge Scale

In Japan, no validated and reliable knowledge scale on HPV and HPV vaccination currently exits. Internationally, however, several such scales have been developed [35,36,37,38,39], among which the HPV Knowledge Scale (HPV-KS), created by Waller et al. in 2013, has been widely used in studies HPV-related research [38].

The original HPV-KS consists of three subscales: general HPV knowledge (16 items), HPV testing knowledge (6 items), and HPV vaccination knowledge (7 items), totaling 29 items. Each item uses a three-option response format: “True,” “False,” or “Don’t know.” Correct responses are scored as 1 point, while incorrect or “Don’t know” answers receive 0 points. Higher total scores indicate greater knowledge. The scale was developed and validated using data from a survey of 2409 men and women in the United Kingdom, the United States, and Australia, and has demonstrated high reliability and validity.

In Japan, cervical cancer screening using cytology is recommended once every two years for individuals aged 20 years and older, but the inclusion of routine HPV testing remains under discussion [40]. For this study, we adopted two of the three original subscales: general HPV knowledge and HPV vaccination knowledge. The general HPV knowledge subscale showed a Cronbach’s alpha of 0.849. The HPV vaccination knowledge subscale had a relatively low Cronbach’s alpha of 0.561; although removing the item “HPV can cause HIV/AIDS” might improve the scale’s internal consistency, it was retained due to its perceived importance in previous research [41], and we chose to retain it as well. Permission to use only the two subscales was obtained from the original author (Table 1).

### 2.2. Additional Questions

To address issues specific to HPV vaccination in Japan, particularly long-standing public concerns over vaccine side effects, three additional items (Q27–Q29) were added. To highlight the risks of HPV-related diseases in both males and females beyond genital warts, three more items (Q17–Q19) were included. In total, six new items were incorporated, resulting in a final 29-item scale (Table 1).

### 2.3. Translation

The original scale was translated from English into Japanese by a university faculty member specializing in nursing and midwifery along with graduate students proficient in English. Care was taken to preserve the original meaning, while ensuring the language was natural and easily understandable for Japanese respondents. A panel consisting of the principal investigator, research collaborators (including an obstetrician-gynecologist), and graduate students reviewed the translated version for content validity. Based on their feedback, a preliminary draft was created and further refined through discussions between the principal investigator and research collaborator.

Special attention was given to five items from the original scale: “Girls who have the HPV vaccine do not need a smear test when they are older,” “HPV can cause genital warts,” “HPV can be cured with antibiotics,” “One of the HPV vaccines offers protection against genital warts,” and “The HPV vaccine requires three doses.” These were carefully adjusted to ensure natural and culturally appropriate wording in Japanese. Additionally, the item “HPV can be passed on by genital skin-to-skin contact,” was translated more broadly to avoid overly narrow interpretations of the terms “genital” or “skin.”

### 2.4. Back-Translation

To ensure the accuracy and conceptual equivalence of the translated scale, a back-translation was conducted by Crimson Interactive Japan, Inc. (Tokyo, Japan) [42], a professional academic English editing service. In this process, a native Japanese translator who had no access to the original English version, translated the Japanese version back into English. A third-party reviewer then compared the original and back-translated versions, focusing on terminology, expressions, and subtle nuances. This evaluation confirmed that the translated scale accurately conveyed the concepts and meanings of the original. Additionally, the principal investigator and research collaborators reviewed both versions to verify the appropriateness of the wording for each item.

### 2.5. Cognitive Debriefing

To confirm the clarity and comprehensibility of the translated scale, cognitive debriefing was conducted with eleven graduate students. Based on their feedback, any discrepancies between the original English items and back-translation were carefully reviewed, and the appropriateness of the Japanese wording was confirmed. This process ensured both linguistic and cultural appropriateness of the Japanese version.

### 2.6. Participants

Between April and August 2024, the study targeted parents or guardians of all students enrolled at six junior high schools in two cities and two towns in Prefecture A, Japan, where research cooperation was obtained. As of 1 April 2024, these students had received HPV vaccination vouchers distributed by the end of March, making them eligible for publicly funded vaccination. This consistent eligibility formed the basis for selecting their parents or guardians as the study population.

We chose parents and guardians because they play a crucial role in the decision-making process regarding their children’s HPV vaccination. In addition, compared with adolescents of vaccination age, parents and guardians are more likely to provide consistent and reliable responses to the questionnaire. Although previous studies have primarily focused on mothers, research that includes parents or guardians more broadly is limited. This study therefore defined the target population as “parents or guardians.”

### 2.7. Study Procedure

An anonymous web-based questionnaire survey was employed. After obtaining approval from the principals of each school, a research information sheet containing a Quick Response (QR) code and URL for the survey was distributed to students by their homeroom teachers, with instructions to deliver it to their parents or guardians. For parents or guardians with multiple children enrolled in junior high school, the instructions specified that responses should be based on their eldest child.

Participants who consented to participate in the test–retest reliability assessment received a follow-up questionnaire via email two weeks after completing the initial survey.

The web-based questionnaire was developed and administered using Google Forms, a form-creation tool provided by Google Cloud Japan LLC (Tokyo, Japan) [43].

### 2.8. Data Analysis

For each item on the scale, the response options were “True,” “False,” and “Don’t know,” following the original version. Correct answers were scored as 1, while incorrect or “Don’t know” responses were scored as 0, with higher scores indicating greater HPV knowledge. Parallel and confirmatory factor analyses were conducted using R software, Version 4.4.1 (R Core Team, Vienna, Austria), whereas all other analyses, including descriptive statistics, reliability analyses, and exploratory factor analyses, were performed using IBM^®^ Statistical Package for the Social Sciences (SPSS) Statistics for Windows, version 28.0 (IBM Corp., Armonk, NY, USA). A *p*-value of less than 0.05 was considered statistically significant.

### 2.9. Validity Assessment of the Scale

To assess the sampling adequacy for factor analysis, the Kaiser–Meyer–Olkin (KMO) test and Bartlett’s test of sphericity were conducted. Construct validity, specifically factor validity, was examined through exploratory factor analysis (EFA) with the maximum likelihood method and Promax rotation. The number of factors was determined based on Guttman’s criterion and parallel analysis [44].

Items were retained if they had a factor loading of ≥0.40 on a single factor and no cross-loadings of ≥0.40 on multiple factors. Items with communality values ≥ 0.16 were also retained.

Confirmatory factor analysis (CFA) was subsequently conducted to validate the factor structure derived from the EFA. Model fit was assessed using the Goodness-of-Fit Index (GFI), Adjusted Goodness-of-Fit Index (AGFI), Comparative Fit Index (CFI), and Root Mean Square Error of Approximation (RMSEA).

### 2.10. Reliability Assessment of the Scale

Internal consistency was assessed using Cronbach’s alpha coefficient. Test–retest reliability, reflecting the scale’s stability over time, was examined by calculating Spearman’s rank correlation coefficient between total scores from the first and second administrations. Item-total correlations were also to assess the relationship between each item and the overall score.

### 2.11. Ethical Considerations

For both the initial and follow-up surveys, a research information sheet was provided on the webpage hosting the online questionnaire. Participants were instructed to read the information carefully before beginning the survey and were free to decline participation. This approach ensured informed consent.

The study was approved by the Ethics Committee of Kagawa University Faculty of Medicine (Approval Number: 2024-009).

## 3. Results

A total of 2828 study information sheets were distributed, and 851 responses were received (response rate, 30.1%). Of these, 793 responses (93.2%) were valid. The average age of participants was 44.2 years. The gender distribution of participants’ children was 401 girls (50.6%) and 392 boys (49.4%). Of the 793 respondents, 93.1% were mothers.

### 3.1. Validity Analyses

Guttman’s criterion and parallel analysis suggested a five-factor structure. Of the initial 29 items, 12 were excluded in two rounds of EFA due to factor loadings below 0.40. In the first round, items 1, 2, 4, 10, 11, 12, 14, 24, 25, and 26 were removed, leaving 19 items. In the second round, items 6 and 8 were excluded for the same reason, resulting in 17 items. A final EFA confirmed that all 17 remaining items had factor loadings above 0.40 (Table 2).

A final EFA was then conducted on the remaining 17 items, all of which showed factor loadings above 0.40. Sampling adequacy was confirmed by a KMO value of 0.870, indicating a “very good” level of correlation. Bartlett’s test of sphericity was also significant (χ^2^ = 4833.5, *p* < 0.001), supporting the suitability of the data for factor analysis.

The EFA identified five distinct factors: Factor 1, labeled “Risk of HPV-related diseases and prevention,” including items 18, 17, 19, 13, and 21 with factor loadings ranging from 0.424 to 0.896; Factor 2, “Pathways of HPV transmission and prevention,” with items 3, 5, 7, and 9, and factor loadings between 0.530 and 0.821; Factor 3, “Safety and effectiveness of the HPV vaccine,” including items 28, 27, and 29, with loadings from 0.576 to 0.737; Factor 4, “Limitations of HPV vaccine efficacy,” including items 20, 22, and 23, with loadings from 0.565 to 0.813; and Factor 5, “HPV susceptibility and natural history,” including items 15 and 16, with loadings ranging from 0.710 to 0.790.

CFA was performed to validate the factor model derived from EFA (Figure 1). Model fit indices indicated good fit: GFI = 0.934, AGFI = 0.907, CFI = 0.928, and RMSEA = 0.063. In total, 17 items were retained and included in the final version of the scale.

The model fit indices indicated good fit: GFI = 0.934, AGFI = 0.907, CFI = 0.928, and RMSEA = 0.063. 17 items were retained in the final version of the scale.

### 3.2. Reliability Analyses

The Cronbach’s alpha coefficient for the entire 17-item scale was 0.868, indicating good internal consistency. Among the individual factors, Factor 4 had a slightly lower alpha of 0.688, just below the commonly accepted threshold of 0.70, while the other factors ranged from 0.702 to 0.845, demonstrating acceptable to good consistency.

Item-total correlations ranged from 0.393 to 0.584, all positive, further supporting the scale’s internal reliability.

For test–retest reliability, 67 participants completed both the initial and follow-up surveys. Spearman’s rank correlation coefficient between the two administrations was 0.791 (*p* < 0.001), indicating good temporal stability.

### 3.3. Participants’ Knowledge Score

The mean total score on the J-HPV-KS among participants was 7.60 (SD = 4.24; mini-mum = 0, maximum = 17), with a median of 7.0, an IQR of 4–11, and a range of 0–17. The mean (SD), median, IQR, and range for the subscales were as follows: Factor 1, 1.10 (1.63), median = 0, IQR = 0–2 (range 0–5); Factor 2, 2.73 (1.45), median = 3, IQR = 2–4 (range 0–4); Factor 3, 1.12 (1.14), median = 1, IQR = 0–2 (range 0–3); Factor 4, 2.20 (1.02), median = 3, IQR = 2–3 (range 0–3); and Factor 5, 0.46 (0.74), median = 0, IQR = 0–1 (range 0–2). The floor and ceiling effects for the total score were each 4.9%. A few subscales showed higher floor or ceiling rates, which can be at tributed in part to their narrow scoring ranges.

## 4. Discussion

With the original developer’s consent, we named the newly developed instrument the Japan HPV Knowledge Scale (J-HPV-KS). The J-HPV-KS was developed by adapting the original HPV Knowledge Scale to incorporate elements specific to the Japanese context. The resulting scale consists of 17 items, including 11 from the original and six newly developed items unique to Japan. Notably, one factor consists entirely of newly added items, marking a substantial structural deviation from the original scale. As a result, we decided it would be difficult to maintain consistency with the original and positioned this as a newly developed, distinct scale. We contacted the original developer of the HPV Knowledge Scale to inform them of our study, and the developer kindly responded that no permission was needed and encouraged us to proceed with publication.

The scale features a five-factor structure, with the following factors: “Risk of HPV-related diseases and prevention,” “Pathways of HPV transmission and prevention,” “Safety and effectiveness of the HPV vaccine,” “Limitations of HPV vaccine efficacy,” and “HPV susceptibility and natural history.” In comparison, the Turkish version of the original scale extracted four factors, adapted to Turkey’s cultural context and HPV vaccine policies. We agree with those authors that conducting validity and reliability analyses across different cultural settings promotes broader use of the scale as a standard measurement tool and facilitates cross-cultural comparisons [41]. We believe that such efforts will further advance research toward HPV eradication within each country.

In the current study, Factor 4 showed a Cronbach’s alpha coefficient of 0.688, which is slightly below the commonly accepted threshold of 0.70. However, Cronbach’s alpha highly depends on the number of items, and Factor 4 consisted of only three items. Despite this, the items demonstrated acceptable factor loadings (0.565–0.813) and communalities above 0.30, supporting their contribution to the construct. Moreover, the items were conceptually coherent, addressing misconceptions regarding the limitations of HPV vaccine efficacy. Therefore, Factor 4 was retained in the final scale structure.

The J-HPV-KS provides a more subdivided view of HPV and vaccine knowledge. For HPV, it covers associated diseases, transmission modes, and natural history. For vaccines, it addresses safety, efficacy, and the scope of protection. While its Cronbach’s alpha (0.868) did not reach the Turkish version’s 0.96, it is comparable to the original scale’s 0.838, indicating sufficient internal consistency.

Based on these findings, the scale is valid and reliable for use in contemporary Japanese society. This study did not restrict the gender of children, anticipating potential expansion of HPV vaccination programs to include boys in Japan. Although male vaccination is considered less cost-effective [45], and studies have shown lower antibody titers in young men compared to women [46,47,48], expanding vaccination to boys could maximize public health benefits and help gender stereotypes and misconceptions disproportionately affecting women [49]. Equitable sharing of HPV prevention benefits is essential, and we hope this scale contributes to advancing research toward that goal.

Of the 793 respondents, 93.1% were mothers. This imbalance in parental representation highlights that most guardians participating in such surveys are mothers. Although this study focused on parents and guardians, the Knowledge of HPV Scale is not limited to this population. The scale may be applied to other groups, such as adolescents, young adults, and healthcare providers, and future research should explore its validity and utility in these populations.

The HPV knowledge scale developed in this study may be applied in various practical contexts. For example, it could be implemented in vaccination campaigns to identify groups with limited knowledge and tailor communication strategies that address specific misconceptions. In addition, the scale can be used in educational interventions in schools or community health settings to evaluate baseline knowledge. Furthermore, the scale can guide the selection of priority topics for awareness campaigns and educational materials by identifying items for which misconceptions or insufficient knowledge are most prevalent. Such applications would enhance the scale’s utility and improve HPV vaccine uptake in real-world settings.

This study is not without limitations. First, the generalizability of the findings is limited because the participants were parents of junior high school students from a single prefecture in Japan, participation was voluntary, and most respondents were mothers (93.1%). These factors may have introduced sampling bias and restricted the representativeness of the study population. Second, the Cronbach’s alpha coefficient for Factor 4 of the scale was relatively low (0.688), warranting a cautious interpretation of this subscale. Third, the same dataset was used for both the EFA and CFA. Although this approach is common in scale development studies with sample size constraints, it limits the generalizability of the findings. Fourth, although test–retest reliability was examined with approximately 60 participants, the sample size was relatively small. This limitation may reduce the statistical power and introduce potential bias, such as the limited representativeness of the target population. Therefore, the findings regarding test–retest reliability should be interpreted with caution, and further studies with larger and more diverse samples are warranted.

## 5. Conclusions

The J-HPV-KS, consisting of 17 items, was developed by modifying the original HPV Knowledge Scale. It includes 11 items from the original and 6 newly developed ones tailored to the Japanese context. The scale was confirmed to be valid and reliable for use in the country. It is recommended as a screening tool to assess HPV-related knowledge and may also support research and inform more effective interventions aimed at increasing HPV vaccine uptake. Future research should evaluate the validity and utility of the J-HPV-KS in broader populations, such as adolescents, young adults, and healthcare providers, and examine its application in educational and community health settings to enhance HPV vaccine uptake further.

## Figures and Tables

**Figure 1 healthcare-13-02536-f001:**
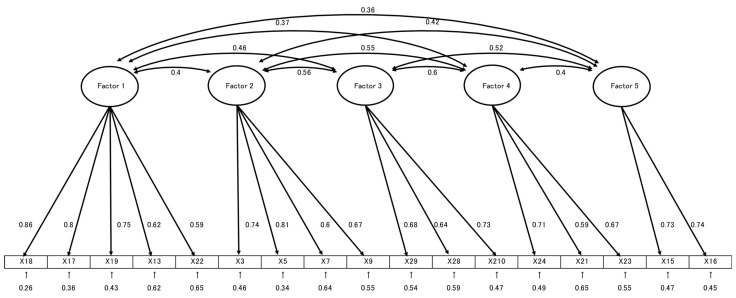
Confirmatory factor analysis of the J-HPV-KS.

**Table 1 healthcare-13-02536-t001:** The 29 Items of the Initial HPV Knowledge Scale (Original and Additional Items).

Item No.	General HPV Knowledge
1	HPV can cause cervical cancer
2	A person could have HPV for many years without knowing it
3	Having many sexual partners increases the risk of getting HPV
4	HPV is very rare (F)
5	HPV can be passed on during sexual intercourse
6	HPV always has visible signs or symptoms (F)
7	Using condoms reduces the risk of getting HPV
8	HPV can cause HIV/AIDS (F)
9	HPV can be passed on by genital skin-to-skin contact
10	Men cannot get HPV (F)
11	Having sex at an early age increases the risk of getting HPV
12	There are many types of HPV
13	HPV can cause genital warts
14	HPV can be cured with antibiotics
15	Most sexually active people will get HPV at some point in their lives
16	HPV usually doesn’t need any treatment
17	HPV can cause anal cancer
18	HPV can cause penile cancer
19	HPV can cause pharyngeal cancer
	**HPV Vaccination Knowledge**
20	Girls who have the HPV vaccine do not need [Pap test/Smear test/Pap smear test] when they are older (F)
21	One of the HPV vaccines offers protection against genital warts
22	Th The HPV vaccines offer protection against all sexually transmitted infections (F)
23	Someone who has had HPV vaccine cannot develop cervical cancers (F)
24	The HPV vaccines offer protection against most cervical cancers
25	HPV vaccines require three doses
26	The HPV vaccines are most effective if given to people who have never had sex
27	The incidence of various symptoms (widespread pain, difficulty moving arms and legs, etc.) observed after HPV vaccination is less than 10 per 10,000 people
28	The causal relationship between the various symptoms observed after HPV vaccination and HPV vaccination has not been proven
29	Scientific verification has confirmed that the benefits of HPV vaccination outweigh the risk of side effects

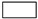
 Original item; 
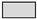
 Added item; (F): False.

**Table 2 healthcare-13-02536-t002:** Exploratory Factor Analysis of the Final 17 Items of the Japan HPV Knowledge Scale (J-HPV-KS).

Items	Factor Loading					Communality
	Factor 1	Factor 2	Factor 3	Factor 4	Factor 5	
Factor 1: Risk of HPV-related diseases and prevention (α: 0.845)						
Item 18	0.896					0.759
Item 17	0.895					0.696
Item 19	0.748					0.571
Item 13	0.485					0.410
Item 21	0.424					0.397
Item 18	0.896					0.759
Factor 2: Pathways of HPV transmission and prevention (α: 0.784)						
Item 3		0.821				0.594
Item 5		0.788				0.656
Item 7		0.643				0.377
Item 9		0.530				0.432
Factor 3: Safety and effectiveness of the HPV vaccine (α: 0.724)						
Item 28			0.737			0.488
Item 27			0.729			0.458
Item 29			0.576			0.458
Factor 4: Limitations of HPV vaccine efficacy (α: 0.688)						
Item 23				0.813		0.585
Item 20				0.588		0.361
Item 22				0.565		0.407
Factor 5: HPV susceptibility and natural history (α: 0.702)						
Item 15					0.790	0.577
Item 16					0.710	0.509
	Factor 1	Factor2	Factor 3	Factor 4	Factor 5	
Factor 1						
Factor 2	0.386					
Factor 3	0.472	0.537				
Factor 4	0.361	0.542	0.558			
Factor 5	0.385	0.416	0.541	0.407		

Factor extraction method: maximum likelihood; rotation converged in six iterations; rotation method: Promax with Kaiser normalization.

## Data Availability

The article’s data will be shared upon reasonable request with the corresponding author. The raw data supporting the conclusions of this study will be made available by the authors upon request. Availability of the J-HPV-KS: This article does not include the Japanese version of the J-HPV-KS. However, the contents of the 17 English items, the answer key, and the scoring method can be identified from Table 1 and Table 2 and the description in Section 2. The full Japanese version is available from the corresponding author upon reasonable request.

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
