# Peer review of "Development and Validation of the Knowledge of Human Papillomavirus Scale in Japan"

_healthcare, 2025, doi:10.3390/healthcare13192536_

Round 1

Reviewer 1 Report

Comments and Suggestions for Authors

Journal: Healthcare (ISSN 2227-9032)

Manuscript ID healthcare- 3811866

Type: Article

Title: Development and Validation of the Knowledge of Human Papillomavirus Scale in Japan

Authors: Ayano Tokuda , Atsuko Shiota , Pasang Wangmo , Kimiko Kawata *

A brief summary

The present study addresses the development and validation of the Japanese version of the HPV Knowledge Scale (J-HPV-KS), adapted from the original HPV Knowledge Scale developed by Jo Waller et al. This research, and its findings, represent a significant contribution to the field by facilitating further investigation into the critically important topic of HPV vaccination within the Japanese population—a context in which such research has not previously been undertaken. This novelty constitutes a major strength and value of the study. The modified scale, as established through the present findings, has demonstrated satisfactory reliability and validity for the specific population of Japanese parents and guardians, thereby supporting its suitability for continued use in research assessing HPV-related knowledge within this target group. While the study is notably timely and addresses an important public health issue, and the manuscript is engaging in certain sections, substantial revisions remain both necessary and advisable. My detailed comments and recommendations for strengthening the manuscript are outlined below:

Line 2-3: It might be advisable to include, in the title of the manuscript, an indication of the target population for whom the scale is intended (i.e., parents and guardians). This would make it immediately clear to readers which specific population is being addressed in the study.

Line 13: The information about the period of time when the research was conducted is missing; please add it.

Line 30-31: Please ensure that all keywords are included and formatted in accordance with MeSH standards;  verify and revise the keywords section as necessary to align with these criteria.

Extensive revisions and additions are required in the Introduction section:

Line 35: Please clarify and add appropriate reference(s) indicating which specific countries are being referred to. Additionally, it would be beneficial to include a few sentences regarding the overall HPV vaccination coverage rates by continent or region, supported by up-to-date references. Furthermore, it would be beneficial to consider adding information on HPV vaccination coverage rates in Japan and its neighboring countries. This contextual information would significantly enhance the manuscript and provide a more comprehensive understanding of both global and regional HPV vaccination efforts.

Line 45: Please clarify why the given percentage is considered inadequate and below target values. For instance, it is important to mention that the World Health Organization (WHO) recommends a vaccination coverage target of at least 90% for girls by age 15 as part of its global strategy to eliminate cervical cancer; and add appropriate reference(s) such as: Beating Cancer Inequalities in the EU: Spotlight on Cancer Prevention and Early Detection, OECD Health Policy Studies, OECD Publishing, Paris, https://doi.org/10.1787/14fdc89a-en or World Health Organization. Global Strategy to Accelerate the Elimination of Cervical Cancer as a Public Health Problem; World Health Organization: Geneva, Switzerland, 2020.).

Line 54: Given that the vaccination of boys is mentioned, please specify which countries provide HPV vaccination for boys as well as the age at which vaccination is administered. I recommend enhancing this section by including recent and authoritative references such as: Kotromanović Šimić, I.; Bilić-Kirin, V.; Miskulin, M.; Kotromanović, D.; Olujić, M.; Kovacevic, J.; Nujić, D.; Pavlovic, N.; Vukoja, I.; Miskulin, I. (2024). The Influence of Health Education on Vaccination Coverage and Knowledge of the School Population Related to Vaccination and Infection Caused by the Human Papillomavirus. Vaccines, 12, 1222 (https://doi.org/10.3390/vaccines12111222), or OECD (2024), Beating Cancer Inequalities in the EU: Spotlight on Cancer Prevention and Early Detection, OECD Health Policy Studies, OECD Publishing, Paris (https://doi.org/10.1787/14fdc89a-en).

Lines 55: Considering that the concept of herd immunity is mentioned, it would be beneficial to reference recent research addressing this phenomenon. Therefore, I recommend enhancing the relevant section by including the following article: Tim J. Palmer, Kimberley Kavanagh, Kate Cuschieri, Ross Cameron, Catriona Graham, Allan Wilson, Kirsty Roy, “Invasive cervical cancer incidence following bivalent human papillomavirus vaccination: a population-based observational study of age at immunization, dose, and deprivation,” JNCI: Journal of the National Cancer Institute, Volume 116, Issue 6, June 2024, Pages 857–865. 

Line 58: If it is possible, please include the approximate price of the vaccine in a widely recognized global currency, such as US dollars or euros, to facilitate clearer understanding of the context for international readers.

Line 60: Could the authors please clarify in greater detail the exact year when HPV vaccination was introduced at the national level in Japan (for both girls and boys)? Additionally, if possible, it would be beneficial to provide a comparative overview, indicating the corresponding years of national HPV vaccination program initiation in neighboring countries or major global nations (also for both girls and boys).

Line 71-72: As in the previous section of the introduction, I kindly request that you add references to support the statements made. This is especially important given the substantial body of literature available that can substantiate the claims presented.

The Materials and Methods section is clearly presented and does not require any significant modifications.

Regarding the Results section, the clarity and overall impact of the manuscript would be substantially enhanced if the authors could included specific data reflecting the level of knowledge obtained on the J-HPV-KS within the study population. This could be presented in the form of tables, figures, or at minimum, a succinct descriptive summary. Although the primary aim of this manuscript is not an analysis of participants’ knowledge per se, providing such information would offer valuable context for interpreting the results and enable readers to better understand the distribution and general knowledge levels among the participants.

The Discussion section requires further strengthen and enhancement in order to provide a more comprehensive interpretation of the findings. Additionally, references 33, 34, and 35 are missing; therefore, these should be included or the reference list revised accordingly.

Line 292-297: A more detailed discussion of the study's limitations and shortcomings would be beneficial.

The conclusion section is well-written and concise; however, it would be beneficial to include recommendations for future research.

Lines 332-432: The formatting of the references is inconsistent and requires careful revision to ensure full compliance with the journal’s author guidelines. Additionally, it is advised, whenever possible, to exclude references older than ten years and replace them with more recent ones.

Reviewer 2 Report

Comments and Suggestions for Authors

It was my pleasure to review this manuscript, which aims to develop and validate a Japanese scale measuring knowledge about the human papillomavirus (HPV) and its vaccine, given the absence of a reliable and validated tool in Japan and the specific context of low vaccination coverage in the country.

With the sole purpose of improving the quality of the manuscript, I offer the following recommendations:

It is suggested to expand the discussion regarding the generalizability of the findings, incorporating the proposal to conduct multicenter or nationwide studies that would allow for the evaluation and confirmation of the scale’s external validity.

Regarding Factor 4, a Cronbach’s alpha coefficient of 0.688 was observed, a value slightly below the commonly accepted reference threshold (≥0.70). In this case, it would be pertinent to provide a solid justification supporting its retention in the final structure of the instrument.

Finally, it is recommended to expand the section dedicated to practical applications by incorporating concrete usage scenarios—such as its implementation in vaccination campaigns or educational interventions—in order to more clearly illustrate its utility and potential impact in real-world contexts.

The manuscript presents a solid methodological development and addresses a real public health need. The scale represents a tool of great potential for educational interventions and epidemiological studies in the Japanese context.

Reviewer 3 Report

Comments and Suggestions for Authors

This manuscript reports the development and psychometric validation of the Japan HPV Knowledge Scale (J-HPV-KS) using responses from 793 parents/guardians of junior-high students. The authors adapted items from Waller et al., added Japan-specific content, reduced 29 items to 17 via EFA, confirmed a five-factor model by CFA (GFI = 0.934, CFI = 0.928, RMSEA = 0.063), and report acceptable internal consistency and two-week test–retest stability (r = 0.791, n = 67).

  1. The authors are encouraged to provide the full list of the final 17 items (Japanese + English back-translation), the answer key, and exactly how the total and any subscale scores are calculated. Include a brief worked scoring example.
  2. Can you please briefly describe how the items went from 29 to 17 (what was dropped and why). A small flow diagram or a supplementary table is recommended.
  3. It looks like the same dataset was used to build and confirm the structure, its often recommended to split the sample (or use simple cross-validation) to confirm the model on a second part of the data.
  4. Please state the software/packages used and how missing answers and “Don’t know” were handled.
  5. Can the authors add mean, SD, median, IQR, and floor/ceiling rates for the total and subscales.
  6. The manuscript would benefit from proofreading by a native English speaker.
  7. Consider depositing the final scale (Japanese + English) as a supplementary file or in a public repository so others can use it.
  8. Please keep JPY amounts but add approximate USD/EUR (and/or GBP) equivalents in parentheses

Reviewer 4 Report

Comments and Suggestions for Authors

Dear Author, 

This work addresses a critical gap in HPV research in Japan and below are my detailed comments to enhance clarity, methodology, and impact: 

The abstract clearly states the goal to develop a validated HPV knowledge scale the importance of focusing on parental knowledge is not clearly stated. Moreover, it’s better to state the Cronbach’s alpha in the abstract section. In the Introduction part, it would be better to strengthen the rationale for focusing on parents and clarify the knowledge Gaps in Japan’s to address the need for a context-specific scale. In the methods and analysis section, clearly specify how discrepancies between original and back-translation were resolved. Also, the heading of 4th factor is "Limitations of HPV vaccine efficacy," while the text calls it "Efficacy against cervical cancer and STDs". In the discussion and conclusion section, it’s unclear to me that how this work would help HPV prevention in Japan and need further discussion. In reference section, some references including No. 10 is not accessible.

Also, there are some main limitations which should be clearly addressed and discussed. The generalizability of this work is questionable as the sampling is from a specific region. The other main issue is about the definition of parents as more than 90% of responders were mothers. The authors may consider excluding the 6.9% of responders (fathers) and see if some parameters (specially the AGFI) may improve. Otherwise, you may keep both gender and discuss the reason behind the low number of male responders. The next issue is related to the sample for test-retest reliability which is small and potential bias should be discussed in details.

Best Regards,

Round 2

Reviewer 1 Report

Comments and Suggestions for Authors

Dear authors, thank you for the corrections and revisions made. I believe that this version of the manuscript is significantly better compared to the previous one. I still think it would be good to change the title in accordance with the earlier suggestions, but I respect your decision to consider this version suitable.